# Sources of Free and Added Sugars and Their Nutritional Impact in Diabetic Patients

Tatiana Fernandes [1], Ana Faria [2,*] and Helena Loureiro [2,*]

1   Centro Hospitalar e Universitário de Coimbra, Praceta Professor Mota Pinto, 3004-561 Coimbra, Portugal
2   Instituto Politécnico de Coimbra, Escola Superior de Tecnologia da Saúde de Coimbra, Rua 5 de Outubro—S. Martinho Bispo, 3046-854 Coimbra, Portugal
*   Correspondence: ana.faria@estescoimbra.pt (A.F.); helenasoaresl@gmail.com (H.L.)

**Abstract:** A high consumption of sugar leads to an increase in caloric intake, which in turn will lead to a higher risk of developing health issues. Foods contain both naturally occurring sugars and added sugars. The World Health Organization recommends that the daily intake of free sugars be below 10% of the total daily energy intake. Food performs a key role in maintaining an adequate glycaemic control in people with diabetes. However, there is a low compliance to dietary recommendations, namely in the amount of sugar intake. This review article aims to assess and compare the intake of various types of sugars in the general population and among individuals with and without a diabetes diagnosis, identify the food sources that contribute to the intake of free and added sugars, and understand their impact on health. Studies performed on the general population found that the consumption of sugar was high, and that children and teens are more likely to exceed the recommended amounts. It was found that diabetics consume less total and added sugar than non-diabetics, as well as a less sugary drinks. Guidelines and public health policy measures aimed at limiting the intake of free and added sugars are needed in order to minimize the consumption of foods high in empty calories.

**Keywords:** sugar intake; free sugars; added sugars; diabetes





## 1. Introduction

In recent years, sugar intake has become a public health concern due to its high consumption and connection to various health issues, such as obesity, cardiovascular diseases, diabetes, metabolic syndrome [1–4], tooth decay, nonalcoholic fatty liver disease, and some cancers [2].

Foods include several different types of sugars that can be naturally occurring or added to food.

Added sugars are sugars used in food processing and/or preparation, such as sucrose, brown sugar, corn syrup, dextrose, fructose, glucose, honey, molasses, inverted sugar, lactose, maltose, and fruit concentrates [5,6]. This kind of sugar excludes naturally occurring sugars in fruits, vegetables, whole-milk dairy products and juices, and/or fruit and vegetable purées.

The term 'free sugars' includes all added sugars and the naturally occurring sugars present in fruits and vegetables in the form of juices or purées. However, it excludes the sugars present in fruits, vegetables, and whole-milk dairy products.

All added sugars are free sugars, and both exclude the sugars naturally present in foods such as in fruits, vegetables, and whole-milk dairy, regardless of whether these are fresh, cooked, or dry [6,7].

Since free and added sugars have similar definitions, the World Health Organization (WHO) uses the term "free sugars" [1].

Total sugars are all the sugars presents in foods derived from any source [2,6].

Excess consumption of calories can lead to weight gain and consequently, obesity and its comorbidities. A high consumption of sugars, namely of added sugars, contributes to

eating "empty" calories and, since they carry no nutrients, in excess they promote weight gain/obesity [1,3,8].

The types of sugar provided by different food sources present important differences in health risks. Eating excess calories, weight gain, diabetes, and tooth decay place free sugars as the primary cause of concern [6]. According to Mela et al. [6], these types of sugars should be the main focus of action in public health activities.

The WHO recommends a decrease in the consumption of free sugars to <10% of the total daily energy intake throughout life in children and adults, and, conditionally, also recommends an intake of <5% [1,6,9,10].

Diabetes is a challenging condition in terms of management and monitoring, as it requires important care by the patient, namely in regards to food. Nutritional therapy is key to an adequate glycaemic control [11–13]. However, studies show that there is a low compliance when it comes to dietary recommendations. According to Asaad et al. [12], eating above 10% of your daily total energy intake (DTEI) in sucrose increases blood sugar and triglycerides in people with type 2 diabetes.

This study aims to evaluate and compare the consumption of the various types of sugar (total, added, and free) in the general population and in people with or without a diabetes diagnosis, to identify food sources that contribute to eating free and added sugars, and to understand their effects on health.

## 2. Sources and Methods

Revision of the published literature between January of 2012 and April of 2022, on the consumption of total, free, and added sugars in people with or without a diabetes diagnosis. The electronic databases used were PubMed and Google Scholar, using a boolean operator (AND) and the keywords "free sugars", "added sugars", "diabetes", and "diabetic population".

The inclusion criteria of the selected articles are: intake of various types of sugars, individuals with and without diabetes, and identification of food sources that provide free and added sugars.

The selection of articles was considered by the date of publication (until 2012), title, abstract, and the full text.

## 3. Results and Discussion

In Europe, the energy intake derived from added sugars was 7.5% to 17%, and it was more significant in children and teens (11% to 17% of the DTEI) [2,14].

In Portugal, according to the 2015–2016 National Food and Physical Activity Survey, the national average intake of simple sugars was 84 g per day, making up 18.5% of the DTEI. As for the consumption of free sugars (7.5% of the DTEI), 24.3% of the Portuguese population exceeded the value recommended by the WHO, and a higher incidence was noted in children and teens. "Table" sugar added to food and beverages (21.4%), sweets (16.7%), and soft drinks (11.9%) were the food groups and subgroups that contributed most to the intake of free sugars [15].

A study performed in Swizerland on adults aged 18 to 75 years showed that the daily totalconsumption of added and free sugars accounted for 19%, 9%, and 11% of total energy intake, respectively. Sweets, drinks (soft drinks), and dairy products (yogurts) were the main food sources of added and free sugars. In younger adults (aged 18 to 29), most of these types of sugars came from soft drinks, while in older adults, sweet products such as honey, jams, cakes, and biscuits took up a more significant share [2].

In the Australian population, the intake of added sugars was found to be 10.8% of the total daily calorie intake. In regards to the WHO guidelines, more than half the sample had a higher intake of free sugars than recommended, and children and teens were more likely to exceed the recommended amount [16].

Between 2017 and 2018, 12.7% of the energy intake consumed by the United States population came from added sugars. This intake was greater in ages between 9 to 18 and lesser

in the elderly (>71 years old). Irrespective of age, ethnicity, or income, sugary drinks and confectionery were the main food sources of added sugars in this population in 2011–2018 [10].

In Latin American countries, the intake of total sugars (20.1%) and added sugars (13.2%) was high. The 15–19 age range was that with the highest exhibited intake. Similarly to other studies, intake decreased with the increase in age [2,4,10].

A study conducted by Liu et al. [17], implemented in the Canadian population in 2015, showed similar results to previous studies. The average daily intake of total sugars was 105.6 g (21.6% of the DTEI), 57.1 g (11.1% of the DTEI) corresponded to added sugars, and 67.1 g (13.3% of the of the DTEI) to the consumption of free sugars. The food groups that most contributed to the intake of sugars in the diet of this population were desserts, sweets, and sugary drinks.

When assessing the consumption of sugars in individuals with diabetes, Wang et al.'s study [11] in the United States of America on Hispanic/Latino adults with and without a diabetes diagnosis, they found that people with diabetes ingested less total sugar (19.1% vs. 21.5% of total energy, *p* = 0.002) and less added sugar (9.8% vs. 12.1% of total energy, *p* < 0.001), as well as exhibiting a lower consumption of sugary drinks (8.8% vs. 11% of total energy, *p* = 0.004).

Asaad et al.'s study [12] assessed the intake of added sugar and the main food sources. According to the WHO guidelines, this sample reached the recommended added sugar intake (8.7 ± 4.8%), with desserts, yogurts, chocolates, breakfast cereal, and cakes being the main food sources of added sugars. The authors of this study found that individuals with diabetes consumed less total sugar compared to non-diabetics, probably because carbohydrates are the macronutrient with the highest impact on blood sugar.

As found by several studies, individuals diagnosed with diabetes have a greater degree of awareness when it comes to adopting healthy lifestyles and practicing healthy eating in order to obtain adequate glycaemic control [11,12,18].

A study conducted in Spain (in the year 2013) observed that diabetic adults consumed on average 24 g of added sugars per day, corresponding to 11% of their daily energy intake, while men and younger individuals (aged 18 to 44) had a greater intake of added sugars than this group. The food sources that contributed to the intake of such sugars were sugar, honey and syrups (19.4%), confectionery and baked goods (19.1%), and sweetened soft drinks (13.4%) [13]. A similar result was identified in a sample of subjects with type 1 diabetes, with an average age of 18, in which the intake of added sugars represented 12.4% of the total caloric intake, from sweetened drinks and sweets/desserts [19].

The majority of studies reported that sugary drinks are one of the food sources that most contributed to the intake of free and added sugars [2,8,10,11,15,19].

Research has identified a link between the intake of sugary drinks and between a higher risk of developing type 2 diabetes, with the added sugar content in drinks being the contributing factor to negative health outcomes [8,18]. Furthermore, individuals who consume this type of drinks have on average a more unhealthy lifestyle—less physical exercise, more tobacco consumption and calories, and a poor quality diet [20].

A study carried out with young people from Jordan identified a positive and significant relationship between body mass index and waist circumference in individuals who consumed drinks with added sugars. This relationship was justified by the high energy intake from these drinks, which resulted in a positive energy balance and increased adiposity [21].

Strategies such as reducing the consumption of these drinks prevents the occurrence of diabetes in the general population and allows for a greater control of this condition in individuals with type 2 diabetes, namely in terms of blood sugar levels, weight gain, and inflammation [22].

## 4. Conclusions

While nutrition aspects for patients with diabetes are defined [23], detailed nutritional evaluations and guidelines on the ill-effects of consuming free and added sugars still remain unexplored.

Guidelines and public health policy measures aimed at limiting the intake of free and added sugars are needed [14], such as an overhaul of added-sugar products (soft drinks, yogurts, dairy desserts, confectionery products) particularly in the amount of sugar, which should be reduced and regulated [2,14], a reduction of serving sizes, a taxation of sugary drinks [2,15], and a clear messaging on minimizing the consumption of foods high in empty calories and in promoting healthy drinks and foods [16].

In regards to food labeling, it is essential that it is suitable and that it clearly identifies free and added sugars in food products [1].

The increased prevalence of diabetes and prediabetes associated with an increase in obesity indicates the need for effective strategies that promote the adoption of healthy eating habits and alternatives to a high intake of free sugars [3].

**Author Contributions:** Conceptualization, T.F., A.F. and H.L.; formal analysis, T.F. and H.L; investigation, T.F. and H.L.; data curation, T.F.; writing—original draft preparation, T.F.; writing—review and editing, T.F.; visualization, T.F., A.F. and H.L. All authors have read and agreed to the published version of the manuscript.

**Funding:** This research received no external funding.

**Institutional Review Board Statement:** Not applicable.

**Informed Consent Statement:** Not applicable.

**Data Availability Statement:** The data presented in this study are available on request from the corresponding author.

**Conflicts of Interest:** The authors declare no conflict of interest.

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
