# Peer review of "Sources of Free and Added Sugars and Their Nutritional Impact in Diabetic Patients"

_diabetology, doi:10.3390/diabetology3040049_

Round 1

Reviewer 1 Report

I thank the journal for giving me an opportunity to review the paper titled “Sources of free and added sugars and their nutritional impact in diabetic patients” by Fernandes et al. This is an overall relevant topic to discuss, where the review highlights sources of free and added sugars and the nutritional impacts it has on patients with diabetes. However, the authors seem to have condensed the information far too much that its lacking essential data, volume, and connectivity needed for a relevant review. There are several changes requested before the paper can be reviewed again for publication.

1.       Daily total energy intake = DTEI and not TDEI. Please correct throughout the manuscript

2.       The manner of performing “source and methods” is not thorough at all. It requires detailed revisions (not just the text section, but expansion of data by including a lot more missing studies). It seems the authors covered just areas in Europe, USA and Australia. What about data from other parts of the world? It makes a reader feel as if all other parts of the world have either no issues with DTEI of free/added sugars and associated effects with diabetes or the authors just limited themselves to not expand on the length and breadth of studies available

3.       Even if this is a “mini review”, the data and information must be organized better to have a flow to the article. Article currently lacks volume. Currently, this is a major criticism that the authors have just thrown in a select set of papers to provide information on the topic. Please carefully expand, restructure and revise.

4.       Line 77: who were the two “independent” observers (list their initials in text)?

5.       Line 107: sudden jump to studies having DTEI and effect on diabetes. The authors may want to include studies with diabetes as a separate subsection (and introduce the general perception of the disease (T1D/T2D) and its clinical relevance related to DTEI)

6.       From lines 107-137, the authors have just mixed-up information that randomly informs readers of T1D, T2D, and diabetes in general. Please divide into specific subsections and expand each of these subsections, mentioning the clinical relevance to each specific disease type and presenting a clinical perspective

7.       The conclusions are well-written and insightful, but expand on the clinical and public relevance.

8.       Grammar checks are mandated across the draft

9.       Some reference papers for the authors’ benefit: PMID: 27882410, PMID: 27900447, PMID: 35380611, PMID: 35314768, PMID: 31405679, PMID: 31083526, PMID: 33084291

Author Response

Thanks for the suggested changes to the amendment.
In fact, this topic does not contain much bibliography, making it difficult to review the topic and prepare the article.
We are open for new suggestions.
Sorry for the delay in responding but for professional reasons it was not possible to submit earlier.

Reviewer 2 Report

Fernandes et.al compared the intake of various types of sugars in the general population and among individuals with and without a diabetes diagnosis. This topic is important for the field especially nowadays the increased sugar intake serves as a crucial risk factor for the development of cardiometabolic disease. Unfortunately, the way the authors summarize these findings is not integrated and straightforward. It would be much better if the authors could showing their findings in one or several plots to show some more "integrated" constitution of sugar and/or added sugar intake in different population across countries. 

Author Response

Thanks for your comment.
We tried to add more complementary information to the article but it isn't an easy topic because there is not much literature that identifies the intake of sugars only in the population with diabetes.

Round 2

Reviewer 1 Report

I thank the authors for considering the review comments and making the changes. The authors are requested to add one sentence (as suggested below) in the conclusion section of the paper for incorporating some clinical relevance, before this paper can be accepted for publication. 

"While nutrition aspects have been well-covered for patients with pancreas pathologies such as type-1 diabetes and chronic pancreatitis (references: PMID: 33002260), detailed nutritional evaluations and guidelines on the ill-effects of free and added sugars in disease context still remain unexplored."

Author Response

Thanks for the sugestion.
I added the information in other words.